# Kinematic Analysis of Water Polo Player in the Vertical Thrust Performance to Determine the Force-Velocity and Power-Velocity Relationships in Water: A Preliminary Study

**DOI:** 10.3390/ijerph18052587

**Published:** 2021-03-05

**Authors:** Giuseppe Annino, Cristian Romagnoli, Andrea Zanela, Giovanni Melchiorri, Valerio Viero, Elvira Padua, Vincenzo Bonaiuto

**Affiliations:** 1Department of Medicine Systems, “Tor Vergata” University of Rome, via Montpellier 1, 00133 Rome, Italy; g_annino@hotmail.com (G.A.); gmelchiorri@libero.it (G.M.); 2Sport Engineering Lab, Department of Industrial Engineering, “Tor Vergata” University of Rome, via del Politecnico 1, 00133 Rome, Italy; cristian.romagnoli2@unibo.it (C.R.); vincenzo.bonaiuto@uniroma2.it (V.B.); 3Department for Life Quality Studies, University of Bologna, 47900 Rimini, Italy; 4Robotics and Artificial Intelligence Lab, ENEA “Casaccia” Research Centre, via Anguillarese, 00301 Rome, Italy; andrea.zanela@enea.it; 5Italian Swimming Federation, Stadio Olimpico Curva Nord, 00135 Rome, Italy; valerio.viero@gmail.com; 6Department of Human Science and Promotion of Quality of Life, San Raffaele Open University of Rome, via di val Cannuta 247, 00166 Rome, Italy

**Keywords:** water polo, biomechanics, video analysis, force-velocity relationship, power-velocity relationship

## Abstract

Background: To date, studies on muscle force and power-velocity (F-v and P-v) relationships performed in water are absent. Aim: The goal of this study is to derive the F-v and P-v regression models of water polo players in water vertical thrust performance at increasing load. Methods: After use of a control object for direct linear transformation, displacement over the water and elapsed time was measured, by using a high-speed 2D-videoanalysis system, on 14 players involved in the study. Results: Intra-operator and player’s performance interclass correlation coefficient (ICC) reliability showed an excellent level of reproducibility for all kinematic and dynamic measurements considered in this study with a coefficient of variation (CV) of less than 4.5%. Results of this study have shown that an exponential force-velocity relationship seems to explain better the propulsive force exerted in the water in lifting increasing loads compared to the linear one, while the power and velocity have been shown to follow a second-order polynomial regression model. Conclusion: Given the accuracy of the video analysis, the high reliability and the specificity of the results, it is pointed out that video analysis can be a valid method to determine force-velocity and power-velocity curves in a specific environment to evaluate the neuromuscular profile of each water polo player.

## 1. Introduction

Water polo is characterized by a complex number of movements: swim with speed changes, faster counterattack actions, frequents changes from horizontal to vertical positions, shots, blocks and fight to gain or maintain the position in water [1]. Most of these actions (handlings, shots, fight) performed at high intensity require a vertical position in water [2]. There are two actions in movement of lower limbs of the water polo player that can be identified: the eggbeater kick (cyclic movement) [3,4] and the breaststroke kick (ballistic movement) [5]. The latter skill, involving maximal lower limbs muscle power, is usually adopted in trunk vertical thrust over the water level to complete the pass, in overall shots and in goalkeeper save actions. Indeed, some studies have found in elite female water polo players a significant correlation between the shot speed and the vertical thrust over the water level performed with breaststroke kick [6,7]. Net of sex differences, it is plausible to consider the vertical thrust one of the main skills also for men’s water polo.

From a biomechanical point of view, the maximal vertical thrust is obtained through the breaststroke kick techniques performed with quick movements on horizontal foot plan in extensive abduction, hip in flexion position and fast flexion-extension of knee [8]. Relative to muscular power and strength of lower limbs, it is common practice to test and condition the water polo players directly in the gravitational environment [1,9] without taking into account the specificity principle of neuromuscular and biomechanical performance that has to be transferred directly to the vertical thrust performed in water [10]. Relative to exercises performed on dry land, some authors showed a poor relationship between ground vertical jump and vertical thrust in water [11,12]. Recently, some authors, using different strength and power training methods performed on dry land or combined (dry-land and in-water) or in water only, showed a positive effect on some of the water polo skills performance with different results related to the method used [12,13,14]. Nevertheless, to date, studies on muscle force and power-velocity (F-v and P-v) relationships performed in water are absent [15]. In fact, the relationship between force and muscular contraction velocity has been determined in athletes to evaluate the dynamic neuromuscular characteristics in isotonic or ballistic conditions [15,16,17]. For this reason, individual power load-based training is difficult be carry out in the water taking into account that this is not specific if performed in a gravitational field.

Usually, in gravitational field, the most used devices for their practical applications in determining the mentioned above curves are linear encoders that, through a derivation process of measured space-time values, are able to calculate force and power parameters in relation to displaced mass [16,18]. Therefore, also taking into account the logistical difficulties in applying whatever isoinertial dynamometer in an aquatic environment, it remains mandatory to find a reliable and non-invasive assessment system. The practical goal of this study has been to verify an easy and reliable method, through a 2-D motion analysis approach, whose validity on kinematic measurements has already been shown [19], to assess the vertical displacement reached over the water level—net of the submerged breaststroke kick technique—and the related derivate kinematic as well as dynamic parameters. Furthermore, this needs to determine the accuracy of the measurement system together with the intra-rater and neuromuscular performance reliability of the assessment method used. In order to obtain in the aquatic environment F-v and P-v relationships like those obtained in gravitational field, it a test protocol was used at increasing loads performing the vertical thrust with a breaststroke technique.

## 2. Materials and Methods

### 2.1. Subjects

Fourteen male sub-elite level water polo players, (age 22.7 ± 5; Body Weight, 72.9 ± 8.2 kg; height 178.9 ± 5.2 cm, Body Mass Index 22.8 ± 2.2 kg/m^2^) participating in the regional championships (Serie C level) organized by the Italian Swimming Federation participated in this study. The body mass and height of the subjects were measured to the nearest 0.5 kg and 0.5 cm, respectively (Seca Beam Balance-Stadiometer, Germany). The players with physical problems (pain or injuries) or with low compliance training were excluded from the study. Written informed consent was obtained from participants (n = 14) before being tested. The research was approved by the Internal Research Board of “Tor Vergata” University of Rome. All procedures were carried out in accordance with the Declaration of Helsinki.

### 2.2. Experimental Design

This study requires the subjects to perform in the water vertical thrust tests at increasing load. This has been applied to the subjects by using the Water polo Overload Test/Training (WOT) [1] equipment, shown in Figure 1, that consists of a harness made of belts that are worn by the player and a load which can be fastened to its lower extremity, and does not interfere in any way with the legs’ movements.

The subjects, once they reached the position between the posts, spent a few seconds floating with the eggbeater kick technique to achieve and to maintain the optimal start position keeping the acromion at the same water level as before to perform, by using the breaststroke kick technique, an explosive boost to raise vertically the body as high as possible. In addition, to avoid any coordinative influence, the subjects held their upper limbs to their shoulders during the performance. Then, wearing the WOT system, they started to perform the increasing load test starting from free load condition which represents the reference trial for the test specificity. The player performed vertical thrusts increasing the load by 5 kg at each step (5, 10, 15, 20, 25 kg) where the 25 kg was the maximum load vertically raised at the limit of the buoyancy. The best trial of three measurements in terms of displacement and verticality at each increased load performance was selected for statistical analysis. Each subject completed raised load test with almost three-minute rest time between the trials enough to recovery from single boost performance.

In order, to determine day-to-day reliability, the subjects underwent the same protocol after two rest days. The measurements were performed in the same swimming pool where the subjects usually train with the water temperature of 29 °C, pH of 7.2–7.6, and an environmental temperature that ranges between 24–26 °C at a humidity of 75%. These parameters remained unchanged in both the test days. One week before the test administration, the subjects performed some simulations for familiarization with the equipment. On the first test day, the anthropometric data were recorded and the subjects performed, after a warm-up, an incremental loads protocol test with the WOT. The warm-up exercises were completed in 15 min and consisted general to specific skills performed with a progressive increase intensity.

The subject, to maintain the assigned start position must produce, properly moving his limbs (eggbeater kick), an upward floating force (*F_ek_*) equal to the difference between the sum of his body weight (*W_B_*) plus the weight of the eventual additional load (*W_L_*) and the sum of the respective buoyant forces (*F_bB_* and *F_bL_*) [20]. Equation (1) takes into account a further force (*F_dw_*) related to the power that is wasted to accelerate the water downwards and that does not contribute to the thrust.
(1)$$F_{ek}-F_{dw}=\left(W_{B}-F_{bB}\right)+\left(W_{L}-F_{bL}\right)$$

Considering that the body weight and the buoyant force, respectively (*V_B_* [m^3^] is the volume of the body, *ρ_s_* its density [kg/m^3^], *ρ_w_* the water density (995.96 kg/m^3^ at 29 °C), *g* [m/s^2^] the acceleration due to the gravity and *f* the fraction of the submerged body), are $W_{B}=gV_{B}\rho_{B}$ and $F_{bB}=gfV_{B}\rho_{w}$. The Equation (1) can be written as follow
(2)$$F_{ek}=W_{B}\left(1-f_{0}\frac{\rho_{w}}{\rho_{B}}\right)+\left(W_{L}-F_{bL}\right)+F_{dw}$$
where *f*_0_ is the fraction of the submerged body at starting position (i.e., the volume of the whole body without the head and neck).

Conversely, when the subject has to perform the vertical thrust, he has to provide, by moving his legs with a breaststroke kick, an upward force (*F_bk_*) greater than the sum of the weight force of both load *W_L_* and body *W_B_*, the friction forces *F_fr_*, the buoyant forces *F_b_* and the losses *F_dw_* as reported in the follow expression:(3)$$F_{bk}>\left(W_{B}-F_{bB}+F_{frB}\right)+\left(W_{L}-F_{bL}+F_{frL}\right)+F_{dw}$$
where *F_bB_* and *F_bL_* represent, respectively, the buoyant force of the subject and the load, while *F_frB_* and *F_frL_* are the respective friction forces [20].

The buoyant and friction forces of the load have been evaluated starting from manufacturing features (material, shape and dimensions). Moreover, because the load remains entirely immersed during the whole test, the relative buoyant force always presents the same value. Furthermore, since the friction depends on the displacement velocity, it will be possible, due to the features of the WOT, to use the same value of the vertical thrust velocity computed for the subject.

A different approach is required for the calculation of the same forces for the human body. Indeed, the buoyant force depends on the fraction of the volume of the immersed body that tends to vary during each test because the height of the vertical thrust changes at different loads. Therefore, the accuracy on the computation of such a term depends on a proper estimation of both the volumes of the different parts of the body and its density. In this study, this value of density *ρ_B_* has been simply estimated, for each subject, starting from his weight and height by using the procedure suggested in [21,22]. Moreover, in order to identify the right fraction of the submerged body volume, we used the mean relative percentage values of the volumes of the different segments of the human body [23,24]. Finally, we chose to evaluate the upward force performed with the breaststroke kick considering the buoyant force (*F_bB_*_0_) at the start position only (i.e., when it presents its maximum level) and where the estimation of the immersed body volume shows the minimum error. Consequently, the calculation of the upward force will be underestimated, in the same way, for all the subjects.

In this context, to avoid the difficulties in evaluating the body volumes at different vertical thrust height, we consider this force minus the relative body weight (i.e., $F_{bk}^{\prime }=F_{bk}-W_{B}$). Therefore, neglecting the skin friction drag of the body *F_frB_* and the losses *F_dw_*, the breaststroke force equation to lift the loads becomes
(4)$$F_{bk}^{\prime }=m_{L}\left(a+g\right)-F_{bB0}-F_{bL}+F_{frL}$$
where *a* is value of the acceleration in the thrust and *m_L_* the mass of the load. Thus, the corresponding mechanical power $P_{bk}^{\prime }$ relative to the force exerted with the breaststroke kick during the vertical thrust can be computed as:(5)$$P_{bk}^{\prime }=F_{bk}^{\prime }\cdot v$$

### 2.3. Experimental Procedure

Each trial was recorded at 240 fps (time resolution ~4 ms) with a high-speed camera (Casio Exilim EX-ZR 3700—Japan) that was positioned at a distance of 2.30 m perpendicular at the sagittal plan of the subject in water. To verify the verticality of upper body displacement over the water level, a second camera was placed orthogonally (and at the same distance) to the first one so that the subject lay in the center of view angles of both cameras. No subject that performed a jump too far from his vertical was considered in this study.

The video analysis procedure allows, by processing the acquired videos, the value of the displacement Δ*d* of the vertical jump to be obtained and the time Δ*t* required by the subject to reach the maximum elevation. In detail, the duration of the rising phase of each thrust was obtained by multiplying the frame time by the number of frames between the start of the movement (i.e., the frame where is observed the starting vertical movement) from the buoyance position and the point where the subject reaches the higher position. The starting position was identified where the subject stands stably with the acromion at the water level.

Moreover, the height of each thrust has been evaluated by measuring (in number of pixels) the distance between the position, in the two different frames of start and top position, of the marker placed on the center of the subject’s headgear with respect to the level of the water (Figure 1). The values of mean velocity, force and power were calculated starting from these values while the muscular force and the relative power produced by the subject were computed starting from the maximum displacement reached in the jump by using Equations (4) and (5) respectively. A single operator provided the acquisition of these values, by using specific tools available inside the video analysis software BioMovie *ERGO*© (by Infolabmedia, Italy).

### 2.4. Video Analysis System Accuracy

The size of the images obtained by the camera was 432 × 320 pixels. The calibration factor *K_C_* [pix/cm] has been evaluated by using a 2D-DLT (2D-direct linear transformation) [25] with vertical (post) and horizontal (crossbar) reference objects in the picture placed at the same distance of the subject (i.e., the subject and reference object are in the same calibration plane). Considering as negligible the horizontal displacement of the athletes during vertical jump performance, the post height (86 cm) only was considered for the calculation of the factor *K_c_* that has been evaluated as 0.717 cm/pix.

Moreover, the relative errors (in percentages) of measured displacement ($\varepsilon_{d\%}$) and of measured time ($\varepsilon_{t\%}$) can be evaluated as:(6)$$\varepsilon_{d\%}=\;\frac{\varepsilon_{u\;}K_{C}}{d_{0\mathrm{kg}}}\cdot 100\;$$
where *d*_0kg_ is the average of the height reached by the subjects during the trials at free load and $\varepsilon_{u}$ is the uncertainty error, due to the motion blur [26] in the estimation of the maximum reference point in the vertical displacement.

The time absolute error of the camera can be assumed as equal to a frame time ($\varepsilon_{t}$ = 4 ms) with a negligible jitter considering that the inaccuracy of the internal clock oscillator of the camera can be estimated at less than 0.1 μs. The percentage errors for the forces as well as the power were evaluated according to the usual methodologies for the error propagation [27].

The estimation of the error for the buoyancy force relative to the different subjects was computed to take into account a value equal to 3.5% for the percentage error in the measure of the body volumes ($\varepsilon_{VB\%}$) as reported by [22].

### 2.5. Statistical Analysis

Data in text, tables and figures are expressed as mean ± standard deviation (SD). The Kolmogorov-Smirnov test was used to validate the assumption of normality. Since no significant deviations from normality were detected, the coefficient of variation (CV), interclass correlation coefficients (ICC), standard error of measurement (SEM) and 95% confidence interval (95% CI) were calculated to determine the day-to-day reliability for displacement, time, velocity, acceleration, force and power. Moreover, the ICC was used as assessment test of consistency, repeatability of quantitative measurements made by same operator and to evaluate the athlete’s performance in two different days. Paired *t*-tests with Bonferroni adjustment and the Pearson correlation coefficient (r) were used for between-group comparisons, for test-re-test measurements repeatability and to determine the level of specificity among selected variables of the test. In addition, the effect sizes (ES) were also calculated using Cohen’s d between the pre-test and post-test means [28], where small effect was 0.1, moderate 0.3 and large was 0.5 [29]. The level of statistical significance was set at *p* < 0.05. The IBM-SPSS 20.0 (SPSS, Inc., Chicago, IL, USA) was used for statistical analysis.

## 3. Results

### 3.1. System Accuracy

In order to evaluate the displacement relative error we apply Equation (2) where *d*_(0kg)_ is equal to 68 cm and the uncertainty error $\varepsilon_{u}$ set to 3 pixels, the $\varepsilon_{d\%}$ is equal to 3.11% while the percentage errors for the velocity, acceleration, force and power can be estimated as $\varepsilon_{v\%}=3.37\%$, $\varepsilon_{a\%}=4.25\%$, $\varepsilon_{Fbk\%}=6.89\%$, $\varepsilon_{Pbk\%}=7.58\%$ respectively.

### 3.2. Reliability

Test-retest values of Mean, SD, SEM, ICC, Pearson correlation coefficient (r) and the CV relative to the displacement, velocity, acceleration, force and power performed in the same day and day-to-day are reported in Table 1. The average displacement decreases from 0.69 m (without load) to 0.15 m (load 25 kg), with r ranging from 0.87 at 5 kg to 0.99 at 0, 10 and 25 kg respectively. The thrust performance time (s) decreases at increasing loads with r ranging from 0.86 at 15 kg to 0.99 at 25 kg. Also, the vertical velocity (m/s) decreases as the loads increase with r ranging from 0.95 at 5 kg to 0.99 at 25 kg. Also, the acceleration (m/s^2^) decreases at increase load with r ranging from 0.93 at 20 kg to 0.99 at 5 kg with high correlation values. By contrast with the kinematic parameters, the force increases proportionally to the load ranging from 20.31 N at 5 kg to 304.35 N at 25 kg with high r values ranging from 0.95 at 20 kg to 0.99 at 5 kg. The power increases progressively from 5 kg (r = 0.99) to reach its maximal value at 20 kg (442.70 W with r = 0.94) and then decreases at 25 kg (313.80 W). The ICC of all parameters, expressed in detail in Table 1, showed an excellent level of reproducibility for all measurements. The CV, while remaining low in the kinematic parameters, tends to increase in the dynamic ones reaching its maximum value of 4.32 at the *P_bk_* 10 kg. In addition, the SEM values observed in day-to-day trials are very low in all kinematic parameters considered for each load. The effect size (ES) calculated between pre-test and post-test means, showed a magnitude ranging from small to moderate in all kinematic and dynamic observed parameters (Table 1). The level of statistical significance was set at *p* < 0.05. An IBM-SPSS 20.0 (SPSS, Inc., Chicago, IL, USA) was used for statistical analysis.

### 3.3. Specificity

For the specificity of the method analyzed in this study, the vertical thrust without overloads was considered as a specific water polo skill and, therefore, correlated with the same skill performed at increasing loads. The analysis of correlation between displacement and the force, power and velocity at each load showed a strong correlation with low load (until 20 kg). As the loads increase, these correlations tend to decrease until it becomes not significant at 20 and 25 kg for velocity while for the force and power became non-significant at 25 kg only (Figure 2).

### 3.4. Force-Velocity and Power-Velocity Relationships

Taking in account the means and SD values of force, power and velocity obtained by the measurements showed in Table 1, it was possible to determine a linear relationship between force and velocity and a quadratic curve between power and velocity (Figure 3).

It is worth noting that force and velocity values presented an inverse trend at increasing loads while the power reached the minimum value at 5 kg condition, reached a higher value at 20 kg, and then decreased again at 25 kg load. Both curves, depicted in Figure 2, show the linear and quadratic equation with a high correlation value (r = 0.92 and 0.99 for F-v and P-v curves respectively). Moreover, with a more accurate analysis of the F-v curve, it is interesting to highlight that the values recorded up to 20 kg maintain a linear relationship while at 25 kg the curve tends to assume an exponential like shape (Figure 4).

Therefore, the following exponential equation (Equation (7)) seems to fit better the behavior of the F-v relationship of the incremental loads test (r = 0.99; *p* < 0.001) than the previous linear one (r = 0.92):(7)$$F_{bk}\left(v\right)=F_{0}\;e^{-\;\frac{1}{2}{\left(v-a\right)}^{b}}$$
where *F*_0_ is the maximum value of the force recorded at the lowest value of velocity (constant *a*) of the vertical thrust performed in the test, *v* is the velocity value recorded at each load while the constant *b* allows us to model the growth in the exponential rate.

## 4. Discussion

The results of this study confirm the accuracy of the kinematic parameters measured with the video analysis system. Displacement, time and the calculated parameters as velocity and acceleration showed error values contained below 4.5% in any ballistic performance (breaststroke kick) load conditions, while the dynamic derivate as force and power showed the maximum error below 8%. It needs to be underlined that, for each parameter (measured or calculated), the relative error was less than the mean differences observed among athletes in each load condition performance.

The level of reproducibility of all parameters assessed in this study was very high between the two trials performed in two different days (Table 1) in terms of correlation (r from 0.86 to 0.99) and CV (<4.5%). Thus, the intra-rater reliability on the video analysis system used in this study and the water polo player’s performance provide a consistent result, with an excellent level of ICC, satisfying the basic requirement of any assessment method [16].

Usually, the methods used to determine the F-v and P-v curves of leg extensor muscles are the half-squat weightlifting or jumping test performed at increasing load in gravitational environment. Considering that the specificity represents the most important discriminant criterion of a test [30], it should be emphasized there is scant specificity from biomechanical and neuromuscular points of view between dry half squat, vertical jump and vertical thrust on the water performance [11]. The strong significant correlation showed in this study between the free load vertical displacement and the other kinematic and dynamic parameters obtained at increasing loads (Figure 2), gave to this method a high level of specificity from biomechanical and neuromuscular points of view. Biomechanically, lifting the upper body over the water level means apply a lift force able to counteract the drag force. Indeed, by using the breaststroke kick technique, Sanders [8] showed that the lift forces in the water polo boots are developed through the synergic action of feet where their velocity action is obtained using the anteroposterior and mediolateral directions, followed by the knee extension and trunk straightening from their start angle with respect to the horizontal plane. In this context, squat weightlifting or dry-land jump involves the neuromuscular system in a different biomechanics condition [31]. In addition, Platanou [11] observed no correlation between the vertical thrust on water and the vertical jump on dry land (r = 0.25). From a neuromuscular point of view, in this study the relationship between vertical thrust tends to decrease at increasing load just to become minimal in correspondence of the maximal strength (25 kg) (Figure 3). Furthermore, in the water the muscle contraction does not use the same strategies related to the stretching-shortening cycle and the performance is not characterized neither by the use of elastic energy nor by stretch reflex, typical features of natural gravity movements on developing the ground reaction force. Currently, the methods used to assess the power and strength of the leg extensor and arm muscles are performed in a gravity environment [32]. This study represents the first tentative, in aquatic environment, able to determine the linear F-v and parabolic P-v relationship of lower limb muscles during a vertical thrust performance directly on the water. Both curves maintain the same characteristics of the F-v and P-v relationship observed on an athlete’s performance made in a gravitational environment using a leg or arm extensors isotonic [16,18] or isokinetic devices [33] or ballistic movement [34]. According to Jaric [32], the linear relationship of F-v and consequent parabolic P-v relationship performed in a multi-joint performance showed a strong correlation revealing a high reliability of all the parameters considered in this study as reported in Table 1. Moreover, the second-order polynomial regression of P-v has shown a *P_max_* in correspondence with 20 kg that represents the optimal load averagely expressed by the analyzed subjects. In this context, also the high values of specificity observed with low loads tend to decrease becoming not significant after the *P_max_* load, probably due to a different neuromuscular pattern. In fact, according to the motor unit size recruitment principle of Henneman [35], by using an increasing loads protocol, the water polo players exhibited a decreasing heights and muscle contraction velocity on vertical thrust in relation to increasing muscular strength (increased loads) (Figure 3). Although the force and velocity values satisfy the linearity of the relationship, it is worth noting that, as shown in Figure 4, these values recorded at 25 kg tend to lose this linearity. Indeed, it can be presumed that the force exerted by the lower limbs in holding and lifting very heavy loads reached a saturation level (plateau) without ever reaching the maximum isometric force, which is impossible to obtain in a water environment, as instead it is observed in a gravitational field. In this condition, it seems conceivable to consider that the F-v curve could switch its linearity in an exponential like relationship with heavy loads (i.e., when the vertical displacement will become negligible to the further increasing load without muscular force increase). In this case, the eggbeater and breaststroke kick are performed alternatively to maintain buoyancy, as happens in games during the hard attacks and blocks between centre forward and defender. Indeed, it is feasible to assume that the limiting factor of the maximal force exerted by a breaststroke kick is based upon the maximal buoyant force sustained with eggbeater kick performance [36]. In this context, the exponential model represented by Equation (7), with the strongest relationship (r = 0.99) compared with the previous linear curve (r = 0.92; *p* < 0.001), seems to better explain the development of the propulsive force exerted in the water by water polo players with the breaststroke kick technique [8,11]. Furthermore, the P-v curve (Figure 3) is not influenced by the linear or exponential F-v curve maintaining the same parabolic trend. The scant correlation observed between the maximal strength and boots performance shows that *P_max_* load could be considered as a reference load to plan strength or velocity conditioning training in water polo players.

Then, in accordance with the incremental load method for eggbeater kick used by Melchiorri [1] the method for breaststroke kick used in this study, seems to be able to overcome the specificity limits of all strength and power monitoring and training method performed on the land for water polo players.

As a limitation, the use of a more performing camera or a new sensor system able to detect the space-time variations of players on the water, should allow the improvement of such a measure with a consequent reduction of the error. In addition, the indirect assessment of the volumes of the different sections of the human body could lead to a less accurate estimation than the real buoyant force of the body players at the different heights reached during the vertical thrust.

## 5. Conclusions

Considering the accuracy and the reliability recorded between two consecutive trials and two days’ video analysis measurements and the high specificity of the breaststroke kick performed at increasing loads, it is reasonable to consider the validity of this method. Thus, the kinematic assessment of the water polo player performing specific neuromuscular and biomechanical patterns in specific environments could reduce the bias in the assessment and training. In line with these considerations, this easy and practical method could provide coaches and trainers specific indications of the individual linear (especially at light loads) or exponential F-v and quadratic P-v relationships of each water polo player, useful for strength and power monitoring and conditioning to perform directly on the water without time spent in a non-specific regimen. Future studies should be required to verify these preliminary results calculating more accurately the human volumes and densities. Moreover, the same method should be also verified on female water polo players.

## Figures and Tables

**Figure 1 ijerph-18-02587-f001:**
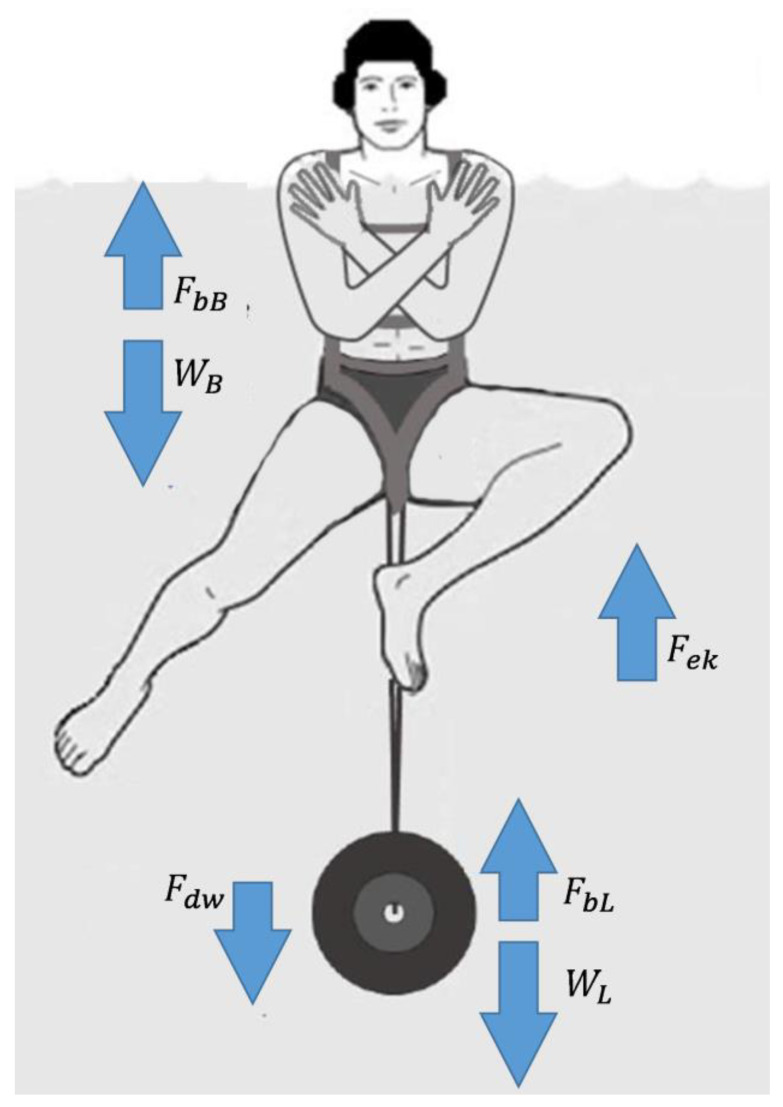
Frontal view of water polo overload test (WOT) conditions and acting forces. Buoyancy force of the subject (*F_bB_*), Body weight of the subject (*W_B_*), eggbeater kick force (*F_ek_*), buoyancy force of the load (*F_bL_*), weight of the load (*W_L_*), force relative to power that is wasted to accelerate water downwards (*F_dw_*).

**Figure 2 ijerph-18-02587-f002:**
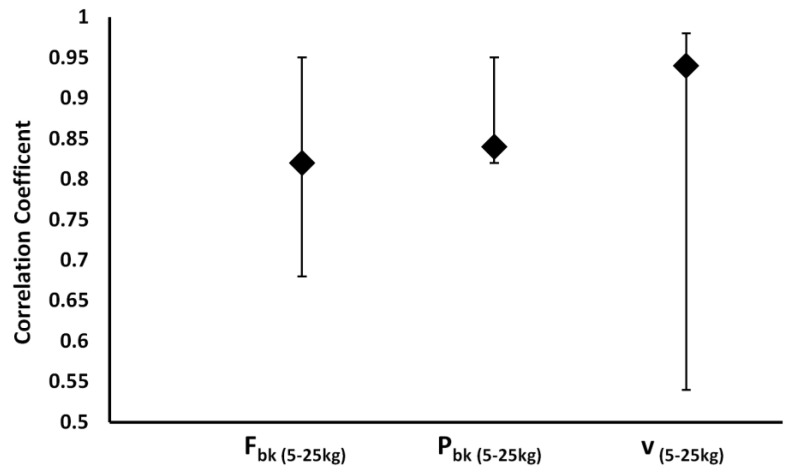
Median correlation coefficients and their ranges obtained comparing the vertical thrust’s height free load with individual *F_bk_*, *P_bk_* and *v* at 5–25 kg.

**Figure 3 ijerph-18-02587-f003:**
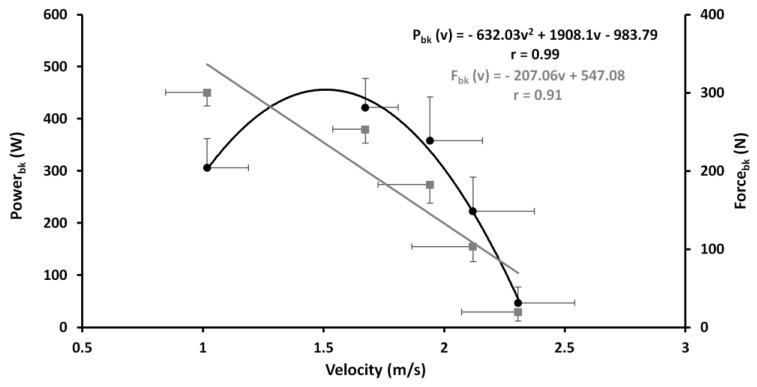
Linear F-v (grey line and squares) and second-order polynomial regression P-v (black line and dots) with the relative regression equations building on vertical thrust performed at incremental loads (from 5 to 25 kg). Both curves are depicted according the average and SD of velocity, force and power at different load as shown in Table 1.

**Figure 4 ijerph-18-02587-f004:**
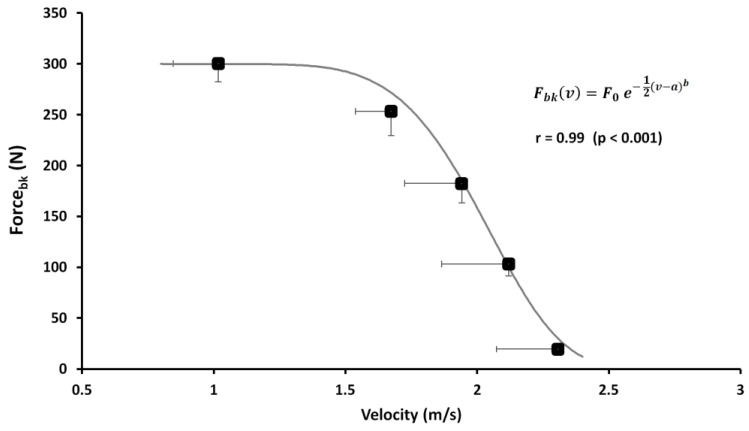
Exponential F-v with the relative regression equations building on vertical thrust performed at incremental loads (from 5 to 25 kg). The curve is depicted according to the average and SD of velocity and force with the different loads as shown in Table 1.

**Table 1 ijerph-18-02587-t001:** Day-to-day repeatability of average displacement (m), time (s), velocity (m/s) and acceleration (m/s^2^), force and power ± Standard Deviation (SD) of the vertical thrust exercise (in water) performed by 14 water polo players with increasing loads, r Pearson correlation coefficient; CV, Coefficient of Variation for repeated measurements; ICC, Interclass Correlation Coefficient; 95% Confidence Interval (CI); SEM, Standard Error of Measurement; and ES, Effect Size, for each load.

Different Day	Day 1	Day 2	r	CV	ICC	95% CI	SEM	ES
Parameters	Mean ± SD	Mean *±* SD						
Displacement (m)								
D_0 kg_	0.69 ± 0.02	0.69 ± 0.01	0.99	0.31	0.99	0.941 to 0.998	0.0013	0.004
D_5 kg_	0.55 ± 0.04	0.57 ± 0.05	0.87	1.61	0.85	0.315 to 0.978	0.0103	0.147
D_10 kg_	0.49 ± 0.05	0.50 ± 0.05	0.99	1.09	0.98	0.910 to 0.998	0.0031	−0.093
D_15 kg_	0.44 ± 0.04	0.43 ± 0.04	0.94	2.43	0.94	0.718 to 0.992	0.0061	0.091
D_20 kg_	0.35 ± 0.03	0.34 ± 0.04	0.97	1.72	0.95	0.595 to 0.993	0.0034	0.219
D_25 kg_	0.15 ± 0.05	0.15 ± 0.05	0.99	0.94	0.99	0.983 to 0.999	0.0013	0.004
Time (s)								
T_0 kg_	0.258 ± 0.009	0.257 ± 0.010	0.95	0.89	0.94	0.713 to 0.992	0.0013	0.134
T_5 kg_	0.244 ± 0.008	0.245 ± 0.010	0.91	1.28	0.94	0.648 to 0.991	0.0018	−0.173
T_10 kg_	0.235 ± 0.009	0.237 ± 0.013	0.90	1.82	0.93	0.561 to 0.990	0.0025	−0.191
T_15 kg_	0.225 ± 0.010	0.225 ± 0.011	0.86	1.78	0.93	0.494 to 0.991	0.0023	0.017
T_20 kg_	0.205 ± 0.008	0.206 ± 0.011	0.96	1.31	0.93	0.627 to 0.991	0.0015	−0.015
T_25 kg_	0.141 ± 0.023	0.143 ± 0.023	0.99	0.73	0.99	0.878 to 0.999	0.0008	−0.075
Velocity (m/s)								
V_0 kg_	2.68 ± 0.17	2.69 ± 0.17	0.98	0.90	0.98	0.891 to 0.997	0.0013	−0.062
V_5 kg_	2.30 ± 0.20	2.30 ± 0.23	0.95	2.13	0.95	0.733 to 0.994	0.0283	−0.023
V_10 kg_	2.11 ± 0.23	2.12 ± 0.25	0.98	1.41	0.98	0.916 to 0.998	0.0173	−0.022
V_15 kg_	1.95 ± 0.18	1.94 ± 0.21	0.97	1.98	0.96	0.808 to 0.995	0.0222	0.066
V_20 kg_	1.71 ± 0.14	1.65 ± 0.12	0.96	1.63	0.93	0.403 to 0.990	0.0159	0.279
V_25 kg_	1.03 ± 0.17	1.02 ± 0.17	0.99	1.38	0.98	0.903 to 0.999	0.0122	0.073
Acceleration (m/s^2^)								
Acc_0 kg_	10.41 ± 1.04	10.51 ± 1.11	0.97	1.72	0.97	0.843 to 0.996	0.1041	−0.094
Acc_5 kg_	9.46 ± 1.09	9.43 ± 1.23	0.99	1.53	0.98	0.912 to 0.998	0.0839	0.025
Acc_10 kg_	8.99 ± 1.21	8.96 ± 1.40	0.96	3.11	0.96	0.752 to 0.994	0.1616	0.024
Acc_15 kg_	8.69 ± 0.95	8.66 ± 0.95	0.97	2.94	0.95	0.712 to 0.993	0.1472	0.031
Acc_20 kg_	8.34 ± 0.78	8.15 ± 0.79	0.93	2.37	0.92	0.552 to 0.988	0.1132	0.241
Acc_25 kg_	7.33 ± 0.46	7.15 ± 0.59	0.94	1.82	0.89	0.234 to 0.992	0.1061	0.323
Force *_Bk_* (N)								
F_5 kg_	20.31 ± 11.39	19.63 ± 11.74	0.99	2.37	0.99	0.976 to 0.999	0.3083	0.058
F_10 kg_	103.55 ± 17.01	103.39 ± 14.00	0.98	2.76	0.97	0.809 to 0.997	2.0393	0.009
F_15 kg_	183.20 ± 19.31	182.55 ± 18.17	0.98	2.16	0.97	0.750 to 0.996	2.8066	0.030
F_20 kg_	257.43 ± 18.51	253.36 ± 17.95	0.95	1.53	0.98	0.787 to 0.996	2.1082	0.223
F_25 kg_	304.35 ± 14.59	300.41 ± 17.15	0.96	1.02	0.93	0.431 to 0.995	2.5036	0.248
Power *_Bk_* (W)								
P_5 kg_	48.27 ± 29.35	47.01 ± 30.06	0.99	3.47	0.99	0.978 to 0.999	0.9558	0.042
P_10 kg_	221.99 ± 58.20	222.95 ± 64.92	0.98	4.32	0.97	0.862 to 0.997	5.5571	−0.015
P_15 kg_	361.15 ± 70.34	358.17 ± 84.30	0.97	4.30	0.96	0.784 to 0.995	8.9433	0.038
P_20 kg_	442.70 ± 62.11	421.90 ± 55.86	0.94	3.30	0.89	0.239 to 0.985	8.2565	0.353
P_25 kg_	313.80 ± 55.93	306.06 ± 55.89	0.97	2.47	0.97	0.733 to 0.998	6.5053	0.138

## Data Availability

All study data are included in the present manuscript.

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
