# Peer review of "Kinematic Analysis of Water Polo Player in the Vertical Thrust Performance to Determine the Force-Velocity and Power-Velocity Relationships in Water: A Preliminary Study"

_ijerph, 2021, doi:10.3390/ijerph18052587_

Round 1
Reviewer 1 Report
This study aimed to derive the F-v and P-v regression models of water polo players in water vertical thrust performance at increasing load. I want to congrats authors for the relevant topic. However, figures should be changed suggesting higher resolution. Please, include new figures until the publication.
Author Response
Point 1: This study aimed to derive the F-v and P-v regression models of water polo players in water vertical thrust performance at increasing load. I want to congrats authors for the relevant topic. However, figures should be changed suggesting higher resolution. Please, include new figures until the publication. Response 1: The authors are indebted with the reviewer for his comment to the work. Following his suggestions, the quality of all the figures in the paper has been significantly improved.Reviewer 2 Report
Line 44. Wrong citation order.
Line 98, 266, 273, 293 - Poor quality of figures.
Line 83 – 84. No explanation of the abbreviations „BW” and „BMI”.
Line 110 - 111. „... up to reach the 25 kg of weight increasing the load by 5 kg at each step and observing three minutes of rest time between each of them.”
Is the three-minute recovery time between sets appropriate?
In article Melchiorri* [2015]. „…The test consisted of different trials until exhaustion with increasing overload from 5 to 25 kg, with a recovery of 20 minutes between each trial…”
What are the reasons for such a difference?
*Melchiorri, G., Viero, V., Triossi, T., Tancredi, V., Galvani, C., & Bonifazi, M. (2015). Testing and training of the eggbeater kick movement in water polo: Applicability of a new method. The Journal of Strength & Conditioning Research, 29(10), 2758-2764.
Line 140 – No pattern numbering.
Line 396. If there is no patent, delete it.
Line 412 – 477. References should be described as follows, depending on the type of work – see – https://www.mdpi.com/journal/ijerph/instructions
Line 422 – 423. Publication no. 5 is missing in the text.
Line 449. Incomplete publication data.
It would be good to add the latest publications to the article:
- Kawai, E., Tsunokawa, T., Sakaue, H., & Takagi, H. (2020). Propulsive forces on water polo players’ feet from eggbeater kicking estimated by pressure distribution analysis. Sports Biomechanics, 1-15.
- Martin, M. S., Blanco, F. P., & De Villarreal, E. S. (2021). Effects of Different In-Season Strength Training Methods on Strength Gains and Water Polo Performance. International Journal of Sports Physiology and Performance, 1(aop), 1-10.

Author Response
Point 1.
Line 44. Wrong citation order.
Response 1: the numbering of the citation has been fixed
Point 2:
Line 98, 266, 273, 293 - Poor quality of figures.
Response 2:
the quality of all the figures in the paper has been improved.
Point 3:
Line 83 – 84. No explanation of the abbreviations „BW” and „BMI”.
Response 3:
the acronyms have been substituted with their relative definitions
Point 4:
Line 110 - 111. „... up to reach the 25 kg of weight increasing the load by 5 kg at each step and observing three minutes of rest time between each of them.”Is the three-minute recovery time between sets appropriate?In article Melchiorri* [2015]. „...The test consisted of different trials until exhaustion with increasing overload from 5 to 25 kg, with a recovery of 20 minutes between each trial...”What are the reasons for such a difference? *Melchiorri, G., Viero, V., Triossi, T., Tancredi, V., Galvani, C., & Bonifazi, M. (2015). Testing and training of the eggbeater kick movement in water polo: Applicability of a new method. The Journal of Strength & Conditioning Research, 29(10), 2758-2764.
Response 4:
the features of the exercise that we asked to be performed by the subjects in this paper is very different from that performed in the Melchiorri’s paper. In such paper, they were asked to perform the trial until exhaustions and this is the reason of a larger rest time. The test used in our paper is related to a single boost performance without involving endurance metabolic processes, so almost three minutes of rest time are enough to recovery the anaerobic metabolic process.
Point 5:
Line 140 – No pattern numbering.Response 5: donePoint 6: Line 396. If there is no patent, delete it.
Response 6:
done
Point 7:
Line 412 – 477. References should be described as follows, depending on the type of work – see – https://www.mdpi.com/journal/ijerph/instructions
Response 7:
the style of the references has been modified following the journal instructions
Point 8:
Line 422 – 423. Publication no. 5 is missing in the text.Response 8: the reference of the publicationno. 5 has been added on the paperPoint 9: Line 449. Incomplete publication data.It would be good to add the latest publications to the article:Kawai, E., Tsunokawa, T., Sakaue, H., & Takagi, H. (2020). Propulsive forces on water polo players’ feet from eggbeater kicking estimated by pressure distribution analysis. Sports Biomechanics, 1-15.Martin, M. S., Blanco, F. P., & De Villarreal, E. S. (2021). Effects of Different In-Season Strength Training Methods on Strength Gains and Water Polo Performance. International Journal of Sports Physiology and Performance, 1(aop), 1-10.
Response 9:
the authors thank the reviewer for the suggestion. Both the references to the papers have been now added and commented
Point 10:
Response 10:
All the notes listed in the pdf file have been modified following the suggestions of the reviewer.

Reviewer 3 Report
This paper aimed to assess P-V and F-V relations. However, the authors presented a method wo assess it, this reviewer has some doubts: 1. What’s the need for different loads? 2. Why compare the different loads? 3. What is the final equation to assess P-V and F-V? 4. If dry land evaluations are the gold standard so far, why not compare these results with dry land? The results are not very clear. Some images are not well explained and there are equations impossible to read in the figures. This study is manly focused in lower limbs. However, lower limbs in swimming only contribute about 10% of thrust. That said, the authors may reanalyse the introduction and discussion regarding possible upper limb(s) contribution. Upon that, this manuscript may have several changes. More commentaries: L143-144: Misses reference. L180-186: This is a difficult procedure. The reason why is because during buoyancy the swimmer may vary his vertical position. Upon that, how was determined the first frame? L.204-205: The authors argue that the participants have made the procedure with and without load. However, a load of 25kg is only presented in the methods section. It would be better to explain the load increments in the methods section. Why 25 kg? L.215: The authors have used parametric methods. However, there are only 14 participants and that can be criticised. However, this reviewer recommend to include the effect sizes on this study. L.236: This reviewer believes that "moment" may not be the most accurate term. L.239-240: The authors may support this loads in the methods section. L.235-252: It would be better to present this information in graphics comparing the loads. Results: Figures have too bad quality. L.269: The figure have bad quality and it is hard to read the equations. Upon that, conclusions are hard to make. Moreover, the authors did not explain the equations....Author Response
Point 1.
What’s the need for different loads?
Response 1:
The different loads are required to increase the strength performance values in relation to the velocity change as performed in any dry land tests (see [16] and [32] in reference of the paper).
Point 2.
Why compare the different loads?
Response 2:
The comparison of the results obtained with different loads are used to assess the physiological changes of force and velocity (force increase and velocity decrease at incremental loads). The paper has been amended and more details on this topic have been added
Point 3:
What is the final equation to assess P-V and F-V?
Response 3:
The authors are sorry for the bad quality of the figures mainly due to the MsWord to pdf conversion. Anyway, all the figures have been modified and their quality improved. The equations are shown (and now should be visible) in the figure 3
Point 4:
If dry land evaluations are the gold standard so far, why not compare these results with dry land? The results are not very clear. Some images are not well explained and there are equations impossible to read in the figures. This study is manly focused in lower limbs. However, lower limbs in swimming only contribute about 10% of thrust. That said, the authors may reanalyse the introduction and discussion regarding possible upper limb(s) contribution. Upon that, this manuscript may have several changes.
Response 4:
The aim of this study was to find a force-velocity and power-velocity relationships considering that the comparison with the same curves obtained in dry land would not been able to carry further contributions to this study. This because the forces exerted by breaststroke kick in water is totally different from squat performed in dry land.
Moreover, it is true, as reported by the reviewer, that, in swimming, the contribution of the lower limbs is less than 10% only. Anyway, this is false during the buoyancy (performed by eggbeater kicking technique) and in the vertical thrust in water (performed by breaststroke kicking technique) (see in the reference in the paper [1] [3] [4] [5] [11] [36]). Because the focus of the paper is not swimming but the vertical thrust in water, the study has been focused on the lower limbs only. For this reason, the subjects, as reported in the paper (Line 111), “took their upper limbs stuck on the shoulders during the performance” to avoid any influence of upper limbs in vertical thrust performance.
Point 4:
L143-144: Misses reference.
Response 4:
The references have been added to the paper
Point 5:
L180-186: This is a difficult procedure. The reason why is because during buoyancy the swimmer may vary his vertical position. Upon that, how was determined the first frame?
Response 5:
More details on the procedure for the estimation of first frame related to the start position have been added on the paper.
Point 6:
L.204-205: The authors argue that the participants have made the procedure with and without load. However, a load of 25 kg is only presented in the methods section. It would be better to explain the load increments in the methods section. Why 25 kg?
Response 6:
The 25 kg load has been, for all the subjects involved in this trial, the maximum value where it has been recorded the lowest vertical displacement value. Following the suggestion of the reviewer, more details on this topic have been added to the paper.
Point 7:
L.215: The authors have used parametric methods. However, there are only 14 participants and that can be criticised. However, this reviewer recommend to include the effect sizes on this study.
Response 7:
It has been verified the test for normal distribution with the Kolmogorov-Smirnov test and obtained a positive response, we proceeded with the parametric analysis as is usually done with sport groups that, usually, consist of samples not too wide.
Point 8:
L.236: This reviewer believes that "moment" may not be the most accurate term.
Response 8:
The authors thank the reviewer for the suggestion.
The Pearson Product-Moment Correlation coefficient (also known as Pearson Correlation Coefficient) was wrongly referred in the text. The relative sentences have been now rephrased and now the coefficient is correctly cited.
Point 9:
L.239-240: The authors may support this loads in the methods section.
Response 9:
The section methods has been improved following the reviewer suggestion.
Point 10:
L.235-252: It would be better to present this information in graphics comparing the loads. Results: Figures have too bad quality.
Response 10:
We preferred to resume all the data and the statistic by using a table. The table has been improved and a new column with the Effect Size (ES) data has been added.
The quality of all the figures in the paper has been improved.
Point 11:
L.269: The figure have bad quality and it is hard to read the equations. Upon that, conclusions are hard to make. Moreover, the authors did not explain the equations....
Response 11:
The quality of all the figures in the paper has been improved and more details on the equations are now reported in the paper.

Round 2
Reviewer 1 Report
Now, the manuscript is suitable for publication.
Reviewer 3 Report
The authors have adress this reviwer questions and recommendations.